# Circulation of Fluorescently Labelled Phage in a Murine Model

**DOI:** 10.3390/v13020297

**Published:** 2021-02-14

**Authors:** Zuzanna Kaźmierczak, Joanna Majewska, Magdalena Milczarek, Barbara Owczarek, Krystyna Dąbrowska

**Affiliations:** 1Research and Development Center, Regional Specialist Hospital, Kamieńskiego 73a, 51-154 Wroclaw, Poland; 2Bacteriophage Laboratory, Hirszfeld Institute of Immunology and Experimental Therapy, Polish Academy of Sciences, Rudolfa Weigla 12, 53-114 Wroclaw, Poland; joanna.majewska@hirszfeld.pl (J.M.); barbara.owczarek@hirszfeld.pl (B.O.); krystyna.dabrowska@hirszfeld.pl (K.D.); 3Laboratory of Experimental Anticancer Therapy, Hirszfeld Institute of Immunology and Experimental Therapy, Polish Academy of Sciences, Rudolfa Weigla 12, 53-114 Wroclaw, Poland; magdalena.milczarek@hirszfeld.pl

**Keywords:** molecular imaging, phage circulation, RFP, HAP phage, pharmacokinetics, murine model

## Abstract

Interactions between bacteriophages and mammals strongly affect possible applications of bacteriophages. This has created a need for tools that facilitate studies of phage circulation and deposition in tissues. Here, we propose red fluorescent protein (RFP)-labelled *E. coli* lytic phages as a new tool for the investigation of phage interactions with cells and tissues. The interaction of RFP-labelled phages with living eukaryotic cells (macrophages) was visualized after 20 min of co-incubation. RFP-labeled phages were applied in a murine model of phage circulation in vivo. Phages administered by three different routes (intravenously, orally, rectally) were detected through the course of time. The intravenous route of administration was the most efficient for phage delivery to multiple body compartments: 20 min after administration, virions were detected in lymph nodes, lungs, and liver; 30 min after administration, they were detectable in muscles; and 1 h after administration, phages were detected in spleen and lymph nodes. Oral and rectal administration of RFP-labelled phages allowed for their detection in the gastrointestinal (GI) tract only.

## 1. Introduction

Advances in the therapeutic use of bacteriophages and encouraging results of recent treatment applications make this kind of antimicrobial agent a promising alternative to antibiotics [1]. Bacteriophages (or simply phages) are viruses that exclusively infect bacteria. Interactions between bacteriophages and human or animal organisms play an important role in phage therapy research [2,3,4]. Potential applications of bacteriophages in medicine, as well as general studies of microbiomes in animals and humans, have induced a need for developing tools that facilitate studies of phage circulation and deposition in tissues and cells. The bioavailability of phages in the living system is shaped by many factors. Some of them depend on phage characteristics e.g., phage capsid morphology, stability [5,6,7,8]; however, in the engineered phages, the multivalent architecture of landscape phages displayed proteins [9,10] that may affect the biodistribution of phage clones able to specifically recognise non-bacterial targets, such as neoplastic ones. Other factors affecting the phage fate in the living stystem are related to the application, i.e., route of administration, timespan, dose [11,12]. The multitude of phage types, preparations, and possible treatment schedules makes it very difficult to propose general conclusions on the pharmacokinetics and bioavailability of phages in vivo, with often incomplete or even contradictory observations [4]. A complete view of phage penetration and clearance in organs and tissues is still not available. There are some reports indicating the survival of phage in compartments of organisms, but complete reports are lacking. Thus, effective tools for the detection of bacteriophages inside living cells or in tissues are needed to extend our knowledge on phage circulation (or penetration) in vivo.

The detection of phages in mammalian cells and tissues can be facilitated by the use of fluorescently labelled phages. Molecular imaging is potentially highly useful for the investigation of phages in living systems. Fluorescent proteins are biological markers that are widely used in cell biology research, particularly in the visualization of protein localization and in the monitoring of biological process dynamics. Furnishing a phage with fluorescent proteins has the potential for use in the molecular imaging of phages [13]. Labelled phages can be used to identify phage fate in vivo, e.g., phage translocation and retention in cells, tissues, and organs [14]. Our previous studies revealed the potential of fluorochrome insertion into the phage capsid; phages labelled with green fluorescent protein (GFP) were able to emit light after excitation and were detectable in murine organs (spleen and lymph nodes) [14]. However, the GFP signal in tissues and cells can be concealed by a relatively high background signal (from tissues). Therefore, GFP seems much more applicable in technological studies and in the environment than in human or animal body-derived samples [15,16].

In this article, we propose red fluorescent protein (RFP)-labelled phages as a new tool for the phage detection in living cells and tissues. We show RFP-labelled phages visualized in living cells (phagocytes, human macrophages SC CRL-9855 cells). Furthermore, a RFP-labelled phage was used to identify phage circulation in tissues (in a mouse model) with different routes of phage administration (intravenously, orally, rectally) in the course of time.

## 2. Materials and Methods

### 2.1. Construction of the rfp-hoc Fusion in the Expression Vector

A combination of two systems—Gateway recombination cloning technology (Invitrogen, Life Technologies Corporation, Carlsbad, CA, USA) and a standard restriction/ligation approach—was applied for gene cloning. Expression plasmids were constructed in a three-step method: (i) construction of an entry clone in a non-expression vector, pDONR™221 (a pUC origin and universal M13 sequencing sites, kanamycin resistance) (Invitrogen, Life Technologies Corporation, Carlsbad, CA, USA, http://tools.invitrogen.com/content/sfs/vectors/pdonr221_pdonrzeo_map.pdf) by introduction of the *hoc* gene and the 5′-terminus restriction sites of the gene, (ii) introduction of the *rfp* gene into an entry clone using restriction sites, (iii) transfer of the coding fragment (*rfp-hoc* fusion) to a destination expression vector, pDEST17 (Invitrogen, Life Technologies Corporation, Carlsbad, CA, USA, http://tools.invitrogen.com/content/sfs/vectors/pdest15_map.pdf) for production fusion of the recombinant proteins. The procedure was carried out according to the Gateway technology manual. Both the *hoc* gene and the *rfp* gene were cloned into the vector as PCR (polymerase chain reaction) products. Recombination sites (*hoc*) or restriction sites (*rfp*) that allowed incorporation of the products into the vector were introduced with PCR primers (*hoc* forward primer: GGCAAAGTTTGTACAAAAAAGCAGGCTCCCGGGAAAGAA TTCATGACTT TTACAGTTGA TATAACTC, *hoc* reverse primer: GGGGACCACT TTGTACAAGA AAGCTGGGTC CTATGGATAGGTATAGATGATACC, *rfp* forward primer: GCCCCGGGCCGAAAACCTGTATTTTCAGGGCAGC AGCAGCATGGTGAGCAAGGGCGAGGAGGTC, *rfp* reverse primer: GCGAATTCGCTTGGGGCGCTTGGGGCCTTGTACAGCTCGTCCATGCCGTAC). The high solubility motif APSAPS was added between RFP (red fluorescent protein) and Hoc by a relevant sequence in the *rfp* reverse primer. PCR was performed on a template of T4 phage total DNA (*hoc*) or MDA-MB-231-luc2-tdTomato cell DNA (*rfp*). PCR products of the *hoc* gene were introduced into the pDONR221 using BP Clonase™ II Enzyme Mix (Invitrogen, Life Technologies Corporation, Carlsbad, CA, USA) according to the manufacturer’s instructions. Constructs were verified by automated Sanger sequencing with standard M13 primers using a 3730 DNA Analyzer, Applied Biosystems, Hitachi, DNA Sequencing Kit BigDye Terminator Cycle Sequencing version 1.1 (Oligo, Institute of Biochemistry and Biophysics, Warsaw, Poland). The appropriate clones were used for restriction/ligation with *rfp* PCR products; restriction enzymes: EcoRI and SmaI (FastDigest, Fermentas, Vilnius, Lithuania) and ligase (LigaFast, Promega, Madison, WI, USA) were used. Constructions were verified by automated Sanger sequencing with standard M13 primers using a 3730 DNA Analyzer, Applied Biosystems, Hitachi, DNA Sequencing Kit BigDye Terminator Cycle Sequencing version 1.1 (Oligo, Institute of Biochemistry and Biophysics, Warsaw, Poland). Proper fusions of the genes (*rfp-hoc*) were transferred by recombination to the expression vector pDEST17 (Invitrogen, Life Technologies Corporation, Carlsbad, CA, USA) in the LR reaction, according to the manufacturer’s instructions.

Hoc.RFP expression clones were tested in *E. coli* B834(DE3) F−ompT hsdSB(rB− mB−) gal dcm met (EMD, Darmstadt, Germany). Strains were checked for ability to express the recombinant proteins rfp-hoc as described: bacterial cells were cultured with intensive aeration in LB (Luria–Bertani broth) high salts (10 g/L of NaCl) culture medium (Sigma-Aldrich, Darmstadt, Germany) with appropriate selection antibiotics (ampicillin) at 37 °C and induced with isopropyl β-D-1-thiogalactopyranoside (IPTG) (0.2 mM) in the exponential growth phase as determined by OD600 measurements (usually at OD600 = 0.8). Further expression was conducted overnight at 25 °C. Bacteria were harvested by centrifugation (6000 rpm, 5 min) and analyzed by SDS-PAGE. Fluorescence was measured in a multilabel plate reader (EnSpire, Perkin Elmer, Turku, Finland).

### 2.2. Phage Display

As a platform for displaying foreign peptides on a capsid, HAP1 phage from the IIET (Institute of Immunology and Experimental Therapy, Wrocław, Poland) Microorganisms Collection was used. HAP1 is a T4 phage mutant (American Type Culture Collection, ATCC, Manassas, VA, USA) with a nonsense mutation in the *hoc* gene resulting in an absence of a functional Hoc protein on the phage capsid.

Bacterial cells were transformed with an expression vector containing *rfp-hoc* fusion. Transformed bacteria were grown at 37 °C in Luria–Bertani broth (LB) with ampicillin as a selection antibiotic until an optical density (OD600) of 0.8 was reached. Next, they were transferred to fresh LB medium containing 0.2 mM of the expression inducer IPTG. After induction of protein expression in bacteria, 10^6^ PFU (plaque-forming unit) of HAP1 (in 1 mL volume) was added. Then, the infected culture was incubated at 37 °C for 8 h. Next, the culture was incubated for 3 days at 4 °C. Lysates were filtered by 0.22 µm sterile syringe filters. Phage titer was determined by spot plating technique. Phage modified with RFP by phage display was denominated “RFP.Hoc.HAP1 phage”, and fluorescence was measured using a multilabel plate reader (EnSpire, Perkin Elmer, Turku, Finland).

### 2.3. Purification of Labelled Phage 

RFP.Hoc.HAP1 phages were purified by size-exclusion chromatography as described by Boratyński et al. [17]; the suspension was chromatographed on a Sepharose 4B column (2.5 × 95 cm, eluent 0.063 M phosphate buffer, pH 7.2, flow 0.3 mL min^−1^). Phages were eluted in the highest molecular weight fraction, as confirmed by fraction titration. Phages were filtered by 0.22 µm sterile syringe filters. Phage titer was determined by spot plating technique.

### 2.4. Phage Imaging in Living Cells

SC cells (cell line, human macrophages, ^®^ CRL-9855™ ATCC collection) were incubated at 37 °C with RFP.Hoc.HAP1 phage or HAP1 phage as a negative control for 5, 10, 20, or 40 min (10^9^ PFU per 10^6^ cells), washed with PBS (phosphate-buffered saline) three times and analyzed. The ratio between phages and macrophages was optimized for sufficient signal detection. In case of lower concentration of phages (10^6^–10^8^ PFU per 10^6^ cells), the signal was undetectable. The optimal ratio between RFP-labelled phages and SC cells for microscopic detection was experimentally established: 1000:1 (phages per cells). Imaging was performed in a fluorescent Fully Automated Inverted Research Microscope for Biomedical Research, Leica, excitation: 550 nm, emission: 580 nm.

### 2.5. Ex Vivo Molecular Imaging

C57BL/6 male, 8-week-old mice, weighing 20–25 g were purchased from the Center of Experimental Medicine, Medical University of Bialystok (Białystok, Poland) and maintained under specific pathogen-free (SPF) conditions in the Animal Breeding Centre of the Institute of Immunology and Experimental Therapy (IIET, Wrocław, Poland). Mice 7 days before each experiment were fed forage without autofluorescence EF AIN93M (Ssniff, Soest, Germany). For 24 h before the experiments, mice were on a water–glucose diet, without forage. To compare different routes of administration of phages, mice (five mice per group, n = 5) received preparations in the following way: (a) orally (p.o.)*:* 1 × 10^11^ of phages (RFP.Hoc.HAP1 or control) were applied by oral gavage directly into the lower esophagus using a feeding needle, (b) rectally (p.r.)*:* the colon was washed with PBS twice; then, 1 × 10^11^ of phages (RFP.Hoc.HAP1 or control) were applied rectally using a cannula, (c) intravenously (i.v.): 1 × 10^11^ of phages (RFP.Hoc.HAP1 or control) were injected into the lateral tail vein. Due to obstacles with intravital imaging of phage circulation in living animals (signal not detected because of skin and fur), after the end of the appropriate time interval (in the range from 20 to 90 min in case of i.v. route of administration and 30 to 90 min in case of p.o. and p.r. routes of administration), animals were sacrificed, and the following organs were excised: lungs, heart, liver, spleen, muscle, kidney, lymph nodes, gut, stomach, and prostate. Imaging of organs was performed in In-Vivo MS FX PRO system (Bruker INC., Billerica, MA, USA) or in a Perkin Elmer Reader. Visualization was carried out using the following settings in an In-Vivo MS FX PRO system: for fluorescence capture t = 30 s, binning 4 × 4, f-stop = 2.80, FOV = 200 (Field of View, excitation: 530 nm, emission: 570 nm). Images were analyzed using Bruker MI software (Bruker INC., Billerica, MA, USA). The intensity of the fluorescence signal is presented as the net intensity of the region of interest and expressed in arbitrary units (a.u.).

## 3. Results

T4 phage was engineered with fluorescent labels by in vivo (in bacteria) phage display [14,18,19]. The first step was the fusion of RFP to an external, structural protein of the phage: Hoc protein. This fusion was produced by the expression of *rfp-hoc* gene fusion in the *E. coli* expression system. Effective expression of RFP-Hoc fusion was assessed by standard SDS-PAGE (Appendix A) and by fluorescence detection in RFP-Hoc producing bacterial cells. Red light emission by *E. coli* expressing RFP-Hoc fusion was significantly higher in comparison to control *E. coli* expressing Hoc protein only (*p* = 0.0017) (Figure 1).

*E. coli* strain efficiently expressing RFP-Hoc fusion was further used in phage display. HAP1 phage, a natural mutant of T4 phage without Hoc protein in the capsid, was used as the phage display platform [20]. Bacterial cells were infected with HAP1 phage after the induction of expression of RFP-Hoc fusion. In these conditions, RFP-Hoc fusion (RFP fused to the N-terminal end of Hoc) and new HAP1 virions were produced in bacterial cells at the same time. RFP-Hoc fusions were spontaneously incorporated into phage capsids, forming phage virions with Hoc and RFP (RPF.Hoc.HAP1 phages) (Figure 2).

Then, newly formed RFP.Hoc.HAP1 phages (Figure 3A) were separated from cell debris containing RFP-Hoc fusions not incorporated on the phage capsids by size exclusion chromatography (Figure 4 and Table 1). Fraction containing non-incorporated RFP-Hoc fusions and other proteins (fraction B in Figure 4) contained also 2 × 10^2^ PFU/mL of residual phages (0.0000004% of phage virions in comparison to fraction A containing the major bulk of labelled phages). Transmission electron microscopy (TEM) analysis has not revealed any significant differences between HAP1 and RFP.Hoc.HAP1 morphology (Figure 3).

Subsequently, phage preparations were dialysed against PBS on high cutoff membranes (also capable of removal of residual non-incorporated RFP-Hoc fusions). The final concentration of purified RFP-labelled phage was 5 × 10^11^ PFU per ml. The fluorescence of labelled phages was significantly higher (*p* < 0.001) in comparison to the control (non-labelled phage HAP1), which had been purified with the same procedure (Figure 5).

Highly purified fluorescently labelled phage RFP.Hoc.HAP1 was used to visualize the phagocytosis of phage particles by human macrophages. Fluorescent phages (1 × 10^9^) were incubated with SC cells (*Homo sapiens* macrophages) (1 × 10^6^). After incubation, phages bound to living cells were visualized by fluorescent microscopy (Figure 6). After 20 min of incubation, the fluorescent signal was strong and detectable. However, a further 20 min of incubation resulted in signal loss. This demonstrates the applicability of RFP-labelled phage for phage detection on mammalian cells, with insignificant (close to undetectable) background fluorescence of the cells. This also suggests that phage particle processing by macrophages may happen within a relatively short time (such as 40 min), resulting in the loss of signal.

Highly purified fluorescently labelled phage RFP.Hoc.HAP1 was further used for molecular imaging of the phage circulation in a murine model, with the comparison of different administration routes. In this experiment, the timing was revealed as one of the key factors determining phage detection in tissues. Phages were detectable at 20, 30, and 60 min after administration depending on the administration route. Ninety minutes after inoculation, phages were invisible (see below).

Furthermore, the intravenous route of phage administration seemed to be the most effective for phages to reach tissues and organs. Intravenously injected phages were spreading within the body in the course of time, and they were detectable in lymph nodes (*p* = 0.0013), lungs (*p* = 0.0365), and livers (*p* = 0.0015) 20 min after administration (Figure 7). Thirty minutes after injection, phages reached muscles (*p* = 0.05) (Figure 7).

Sixty minutes after injection, phages were detectable in the lymph nodes (*p* = 0.0349), (Figure 7) and in spleens (*p* = 0.0351) (Figure 7). Phages were not detectable at any time in the prostate or digestive tract (data not shown). These results demonstrate both the dynamics of phage circulation in vivo and the major organs that are efficiently penetrated by phages (liver, spleen, lymph nodes, lungs, muscles), in contrast to those where phage penetration was poor or ineffective (prostate, the interior of the digestive tract).

Furthermore, phage penetration after different routes of administration was assessed. After p.o. administration (by gavage, directly into the lower esophagus using a feeding needle), in the same dose as in an i.v. and p.r administration, phages were detectable in the stomach (Figure 8). In the case of stomach samples, background fluorescence was still detected, but the level of phage fluorescence was significantly higher in the comparison to the control (*p =* 0.017) after 60 min of administration (Figure 8). Phages were not detectable in other organs (lungs, heart, kidneys, spleen, liver, muscle, lymph nodes, prostate).

The fluorescently labelled phage RFP.Hoc.HAP1 was further used in the *per* rectum route of administration, the signal of labelled phages was detectable in the stomach (*p* = 0.0066) (Figure 9) and gut (*p* = 0.014). In the comparison to the *per os* route of administration, the strength of the signal was similar in the stomach, and 20% stronger in the gut when applied rectally. The signal strength was undetectable in other organs: lungs, heart, kidneys, spleen, liver, muscles, lymph nodes, and prostate.

## 4. Discussion

Phage circulation in animal and human bodies is a key parameter in the therapeutic use of phages [21]. However, the repertoire of phage detection methods is still limited, and those available have significant flaws. The most common approach is based on the tissue (or cell) homogenisation and subsequent culturing of the obtained material with a phage-sensitive bacterial host. This method in fact tests for phage ability to infect bacteria, and it is prone to either the inhibitory or boosting activity of a multitude of compounds released from mammalian tissues (or cells) during preparation. It also requires typically overnight culture and the related delays in detection. To some extent, phage plaques can be masked by dense tissue materials, so low phage concentrations (where low dilutions must be used) are poorly detectable and prone to errors.

The development of an effective method for monitoring phage distribution in various cells and tissues still remains a challenge. Here, we propose the novel biotechnological tool RFP-labelled phage (RFP.Hoc.HAP), with a high potential for highly informative pharmacokinetic studies in living organisms. This phage was engineered to present RFP (approximately 27 kDa) on its surface by the phage display method. RFP-labelled phages effectively emitted red light, and they could be detected by fluorescence microscopy. Foreign peptides exposed on phage capsids can influence the distribution of bacteriophages in bodies. Importantly, the overall morphology of phage particles was not affected by the exposition of RFP on the capsid (Figure 3). However, one possible effect that RFP may exert on phage pharmacology cannot be excluded. According to previous studies, phage circulation was not radically changed by protein presentation on capsids, although they may shape the details of phage circulation kinetics [22,23,24] (no data on RFP presentation are available).

The mononuclear phagocyte system contributes to phage clearance. The uptake of bacteriophages by mammalian cells has been reported previously [14,25,26,27,28]. Here, we investigated the interaction of RFP-labelled phages with human macrophages. The fluorescence of single eukaryotic cells, although ubiquitous when imaging, is often considered to be low when compared to the studied object, which is here the labelled phage. We observed that after 20 min of incubation, phages were successfully visualised with phagocytic cells, importantly, with in fact undetectable background fluorescence from the cells. We detected the fluorescent signal coming from the labelled phages; however, we were not able to determine whether phages were bound to the surface of cells or they were inside macrophages. Both possibilities are probable due to the fact that in the case of macrophages, phagocytosis is a dynamic process.

Phages are able to circulate in the body and accumulate in organs and tissues. The largest amount of bacterial viruses after systematic delivery was observed in the liver, which is crucial in the filtering process that removes phages from animal or human bodies [4,29,30]. In our study, as early as 20 min after intravenous phage administration, phage-emitted fluorescence was detectable in the liver, lungs, and lymph nodes. In the course of time, 30 min after application, phages were present in muscles. Sixty min after the intravenous injection of phages, the signal was detectable in lymph nodes and spleen. In contrast, phages were no longer detectable in the liver. This suggests that the presence of Kupffer cells, which are a critical component of the mononuclear phagocytic system, are crucial and very effective for the fast removal of phages in this particular organ [30]. This study also demonstrated that bacteriophages most efficiently disseminated within the body and organs after intravenous administration. After oral and rectal administration, phage detection in organs was poor. On the other hand, only these two delivery routes were effective for phage delivery to the stomach (in case of *per os* phage application) and additionally in the gut (*per os* and *per rectum* route), where they remained detectable after approximately one hour. Thus, the oral administration of phage was the least effective for systemic dissemination, but it allowed for the highest phage concentrations in the digestive tract.

Phage retention in tissues is at best moderate because of the lack of specific receptors for phages on eukaryotic cells. Nevertheless, phage presence in organs after administration is detectable within minutes, which is consistent with other published results [31,32].

Phages are able to circulate and accumulate in organs. Our new model of phage circulation and accumulation in tissues demonstrated the effect of administration route and the dynamics in the course of time. Labelled phages can be easily detected in tissues after dissection by molecular imaging, but there are some limitations in this technology. One of them is the fact that a fluorescent label can be inactivated, in some cases independently on the inactivation of antibacterial activity of phage. However, in short-term measurements, it offers real-time detection and makes it possible to escape tissue inhibitors; thus, fluorescently labelled phages seem to be a highly applicable biotechnological tool to study phage pharmacokinetics. Another problem is the fact that the emission spectra of RFP compete (to some extent) with the autofluorescent signal from animal tissues. Components such as collagen or red blood cells are strongly autofluorescent, making a background noise that needs to be distinguished from the signal emitted by RFP-labelled phage [33,34]. For this reason, in the presented experiments, the background signal in control organs (excised from animals not treated with the RFP phage) was detectable. However, the fluorescent signal from organs after treating animals with RFP-labelled phages was significantly higher. Non-labelled phages do not emit a significant fluorescent signal, as we demonstrated (Appendix A); thus, the only background that should be considered and accommodated in the analysis comes from mammalian tissues. Correlation between intensity of fluorescence and the total amount of RFP phage in the sample presented in Appendix A can also be used for approximations of phage amounts in investigated samples.

Summarising, the phage-labelled technique is faster, cheaper, and less prone to delays and technical problems than the method based on the titration of non-labelled phages in tissue homogenates. In future studies on phage circulation, this technique can be extended to other fluorochromes, and comparative analysis of other groups of phages (e.g., staphylococcal, pseudomonas, T4-like) can be included, offering a comprehensive approach within phage pharmacokinetic studies.

## Figures and Tables

**Figure 1 viruses-13-00297-f001:**
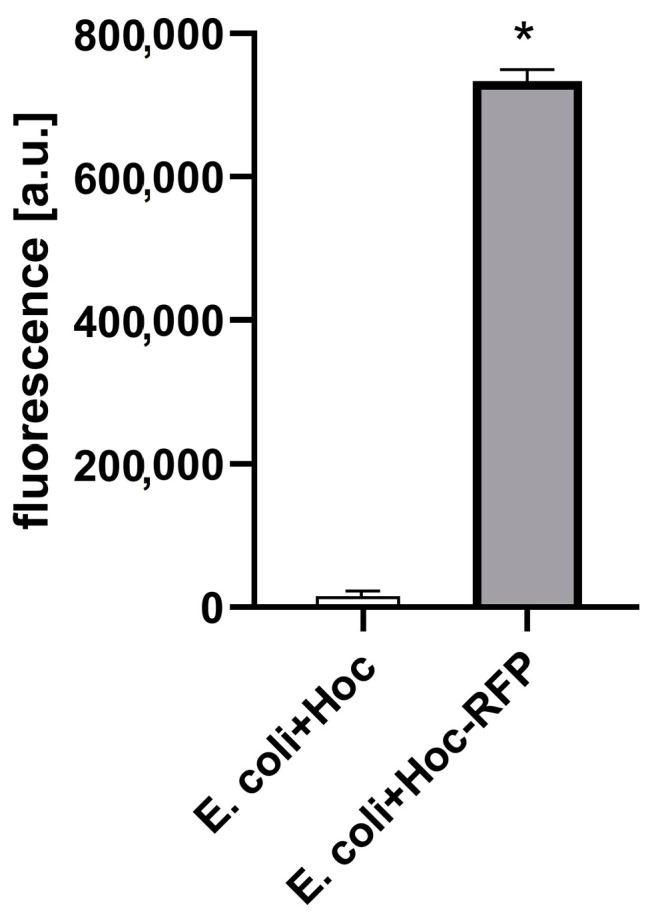
Fluorescence of RFP-Hoc fusion in *E. coli.* Effective fluorescence of the red fluorescent protein (RFP)-Hoc fusion was assessed by comparison of fluorescence in *E. coli* induced for RFP-Hoc production and in control *E. coli* (expressing protein Hoc without RFP) * *p* = 0.0017, unpaired t-test. The error bars represent standard deviation (SD). The experiment was repeated three times with concordant results. One representative experiment is presented in the figure.

**Figure 2 viruses-13-00297-f002:**
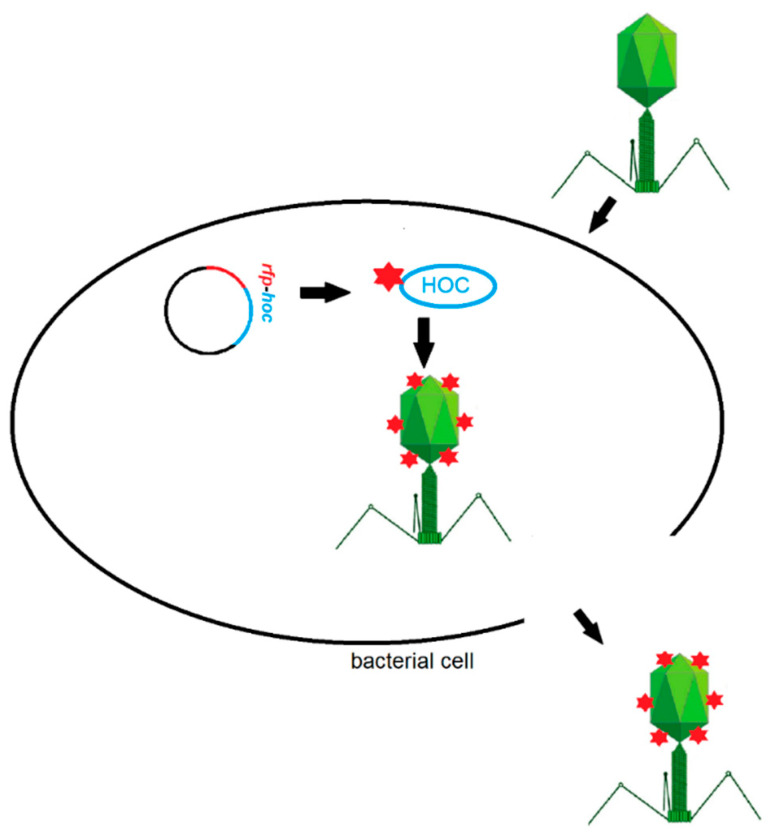
Scheme of red fluorescent protein (RFP) labelling of phages by phage display method.

**Figure 3 viruses-13-00297-f003:**
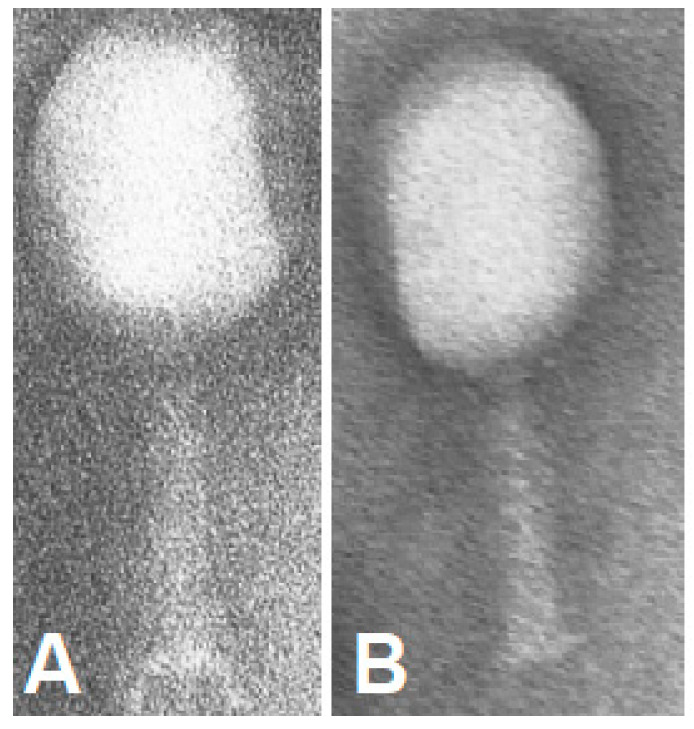
TEM images of RFP.Hoc.HAP1 phage (**A**) and control phage (HAP1) (**B**).

**Figure 4 viruses-13-00297-f004:**
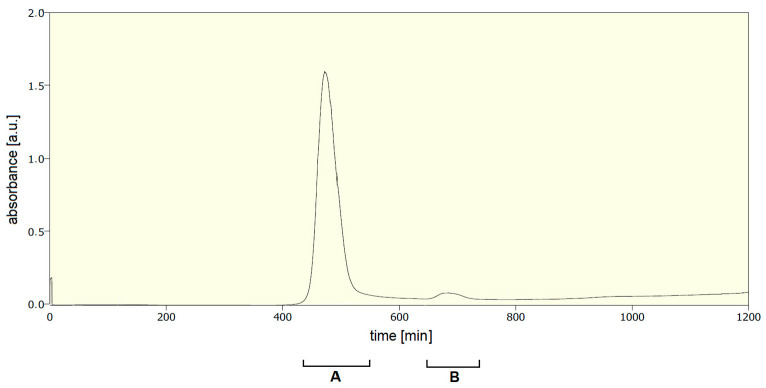
Separation of modified bacteriophages (with RFP-Hoc fusion) by fast protein liquid chromatography. (**A**) The fraction containing bacteriophages. (**B**) The fraction containing non-incorporated RFP-Hoc fusions and other proteins.

**Figure 5 viruses-13-00297-f005:**
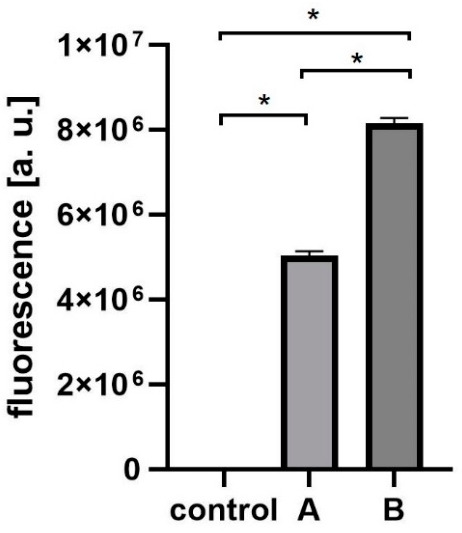
Comparison of fluorescence in RFP-presenting phage (**A**), control phage (control), and fraction of non-incorporated RFP and other proteins (**B**). Effective fluorescence of the purified RFP.Hoc.HAP1 (**A**) phage was compared to the fluorescence of HAP1 phage (without RFP, control), and to (**B**) residues from size exclusion chromatography used to separate non-incorporated RFP-Hoc fusions. The titer of RFP-labelled phage (**A**) and control phage (control) was 5 × 10^11^ PFU (plaque forming units)/mL. Both phages were purified identically: size exclusion chromatography, then dialysed in PBS. * *p* < 0.0001, one-way ANOVA, Brown–Forsythe and Welch test. The error bars represent standard deviation (SD). The experiment was repeated three times with concordant results. One representative experiment is presented in the figure.

**Figure 6 viruses-13-00297-f006:**
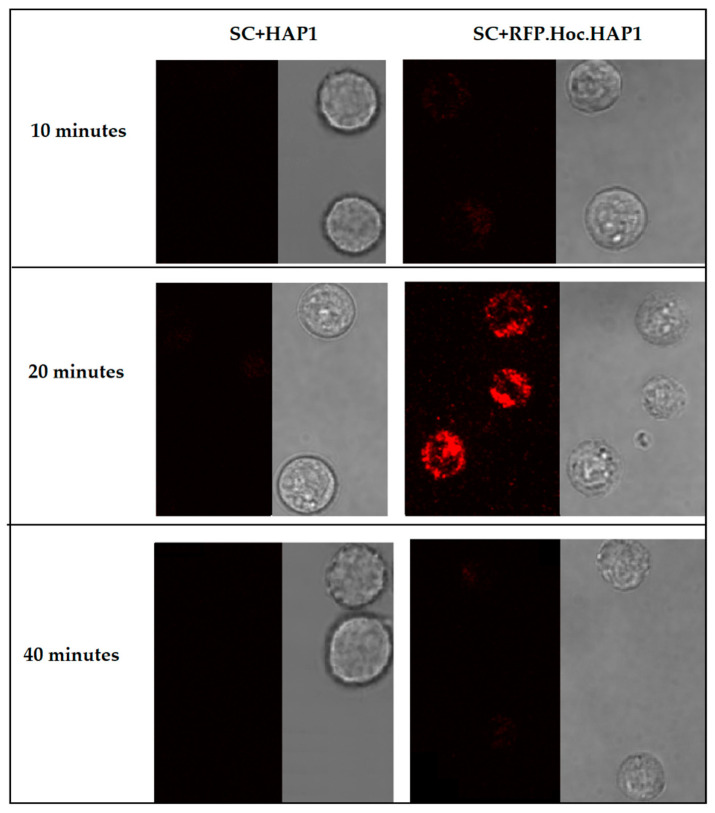
Visualization of RFP-modified bacteriophages in phagocytic cells SC CRL-9855 (SC American Type Culture Collection, (ATCC) ^®^ CRL-9855™ Homo sapiens macrophages). Fluorescence of the human macrophage cell line was visualized in fluorescent microscopy after 10, 20 and 40 min of incubation. SC+HAP1—control, SC cells incubated with non-labelled phage HAP1, SC+RFP.Hoc.HAP1—SC cells incubated with labelled phage RFP.Hoc.HAP1.

**Figure 7 viruses-13-00297-f007:**
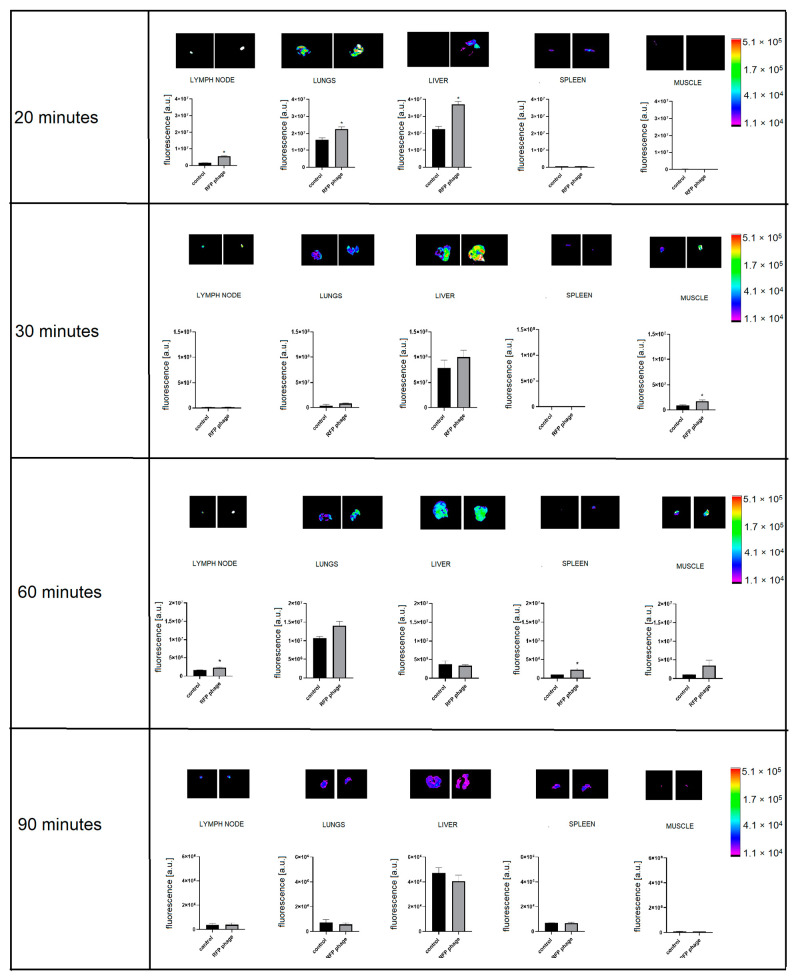
Fluorescence of RFP-displaying phage in murine organs ex vivo after intravenous phage administration. Lymph nodes, lungs, liver, spleen, and muscles were dissected from mice (N = 5) 20, 30, 60, or 90 min after intravenous administration of RFP.Hoc.HAP1 phages (right side) or control (non-labelled phage) (left side); representative images are also shown; * *p* = 0.0013 (lymph nodes after 20 min), *p* = 0.0365 (lungs after 20 min), *p* = 0.0015 (liver after 20 min), *p* = 0.05 (muscle after 30 min), *p* = 0.0349 (lymph nodes after 60 min), *p* = 0.0351 (spleen after 60 min), unpaired t-test. The error bars represent standard deviation (SD). The experiment was repeated twice with concordant results. One representative experiment is presented in the figure.

**Figure 8 viruses-13-00297-f008:**
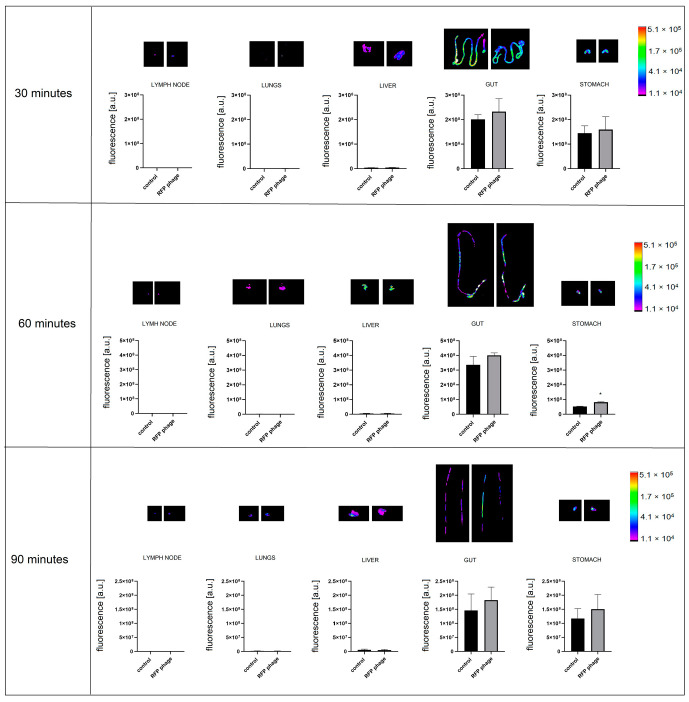
Fluorescence of RFP-displaying phage in murine organs ex vivo after phage administration orally (p.o.). Lymph nodes, lungs, liver, gut, and stomach were dissected from mice (n = 5) 30, 60, and 90 min after p.o. administration of RFP.Hoc.HAP1 phages (right side) or control (non-labelled phage) (left side); representative images are also shown; * *p* = 0.017, unpaired t-test. The error bars represent standard deviation (SD). The experiment was repeated twice with concordant results. One representative experiment is presented in the figure.

**Figure 9 viruses-13-00297-f009:**
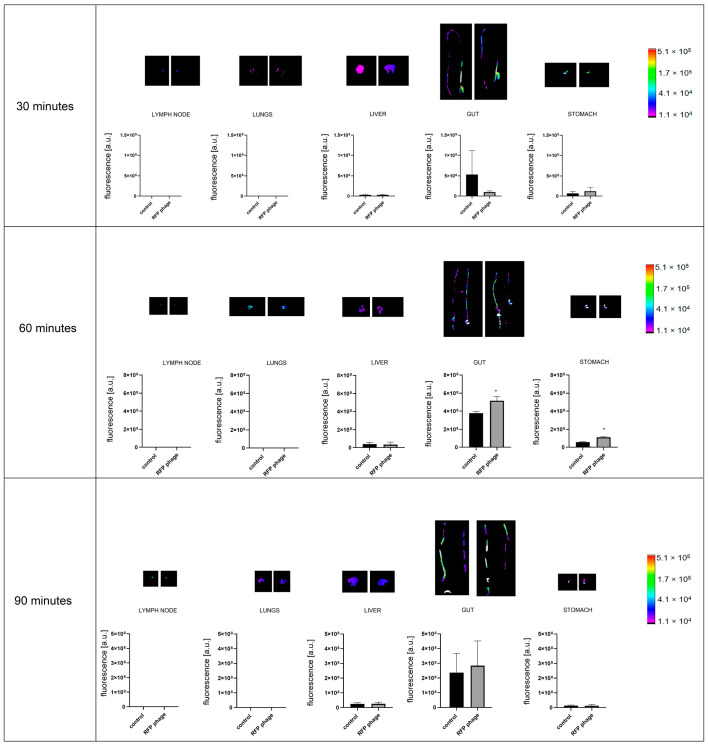
Fluorescence of RFP-displaying phage in murine organs ex vivo after phage administration *per rectum*. Lymph nodes, lungs, liver, gut, and stomach were dissected from mice (n = 5) at 30, 60, and 90 min after rectal administration of RFP.Hoc.HAP1 phages (right side) or control (left side); representative images are also shown. * *p* = 0.0066 (stomach), *p* = 0.014 (gut), unpaired t-test. The error bars represent standard deviation (SD). The experiment was repeated twice with concordant results. One representative experiment is presented in the figure.

**Table 1 viruses-13-00297-t001:** Fluorescence and titer of each fraction after the separation of modified bacteriophages (with RFP-Hoc fusion). Control: nonlabelled phage HAP1, A: fraction after size exclusion chromatography containing RFP.Hoc.HAP1 labelled phage, B: fraction containing non-incorporated RFP-Hoc fusions and other proteins.

Fraction	Titer	Fluorescence [a.u.]
control	5 × 10^11^ PFU/mL	1021
A (labelled phages)	5 × 10^11^ PFU/mL	5,124,785
B (non-incorporated proteins)	2 × 10^2^ PFU/mL	8,274,121

PFU: plaque forming units; a.u.: arbitrary units.

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
