# Peer review of "Circulation of Fluorescently Labelled Phage in a Murine Model"

_viruses, 2021, doi:10.3390/v13020297_

Round 1

Reviewer 1 Report

In this manuscript, the Authors propose a T4 phage mutant labelled with a red fluorescent protein (RFP) to investigate the phage pharmacokinetics in mouse model. Although the Authors show great expertise about the subject matter, the manuscript presents a strong homology with another work previously published by the same authors (https://doi.org/10.4161/bact.28364) and has important issues that need to be addressed before publication.

Major revision:

  • The title should be revised, as well as the sentence "investigation of phage pharmacokinetic in cells" in the end of the introduction.I believe that the Authors did not conduct pharmacokinetic studies on single cells (see https://doi.org/10.1016/j.drudis.2015.05.011), but we did "phage imaging" on the SC ATCC ® CRL cell line -9855 ™. The pharmacokinetic study concerns only tissues, evaluated by "ex vivo molecular imaging"
  • In “Some of them depend on phage characteristics e.g. phage capsid morphology”, the Author should also include references about multivalent architecture of landscape phage displayed proteins (some examples https://doi.org/10.3390/v11110988, https://doi.org/10.1093/nar/gkaa1279) which affect the biodistribution of  phage clones able to specifically recognize non-bacterial targets, such as neoplastic ones.
  • In subsection “Phage imaging in living cells” of Materials and Methods, the Authors use a phage titer per cell (109 pfu per 106 cells) higher than that of their previous works (106 pfu per 106 cells, https://doi.org/10.4161/bact.28364). What is the reason for the choice? Does it depend on the different cell line?  
  • What is the significance of Figure 3? Is it possible to see the RFP on phage capsid?
  • In table 4, the fraction “B (non-incorporated proteins)” is quantized as 2 × 102 pfu/ml. Does this fraction contain residual phages? Phage proteins, able to forming plaques, are present in fraction B? Specify this in the text and consequently change the caption of Figure 5 “…and to (B) a non-phage fraction…”
  • After the purification of the fluorescently labelled phage RFP.Hoc.HAP1 by fast protein liquid chromatography, the Authors should have constructed a calibration curve between phage titer and intensity of the fluorescence, in order to better interpret the experimental data obtained from ex vivo molecular imaging.
  • The Authors performed phage imaging in living cells at several times, namely 5, 10, 20, or 40 min (see Materials and Methods). Consequently, the figure 6 should show a composition of all fluorescence images (also including those where no fluorescence is observed) at different times for both control and Hoc.HAP1 phage.
  • Similarly, results from ex vivo molecular imaging should also be organized in a totally different way. For each type of administration (i.v., p.o. and p.r.), the Authors should mount all fluorescent images (also including those where no fluorescence is observed) from a single tissue at different times (in this case 20, 30, 60, 90 and 120 minutes). As a reader, I have found it difficult to read the results.

Minor revision:

  • The abbreviations should be written in full when they first appear in the text (eg PBS, PFU). This is necessary for readers with other expertise
  • Also the alternate use of phage and bacteriophage terminology could confuse the reader with other terminologies (eg macrophage).In that case, a sentence such as "bacteriophage (or simply phage) is a virus that exclusively infects bacteria" should be added in the introduction.
  • Labelled and Analysed are the preferred spelling in British English.
  • In subsection “Phage imaging in living cells” of Materials and Methods, references are missing for Boratyński et al. and for routine test dilution (RTD).
  • In subsection “Phage imaging in living cells” of Materials and Methods, the number ® CRL-9855™ to ATCC collection.
  • Minutes or min? Hours or h?
  • The text should be fully revised using the guidelines for authors.I also suggest using the word template in https://www.mdpi.com/journal/viruses/instructions

Reviewer 2 Report

Development of effective methods for monitoring distribution of bacteriophages in various organs of animals and humans is a very important subject. Thus, I have enthusiastically started to read this paper. Unfortunatelly, although the idea to construct a fluorescent phage which can be easily detected in animal organs was very interesting, my enthusiasm dropped after seing shortcomes of this method. Nevertheless, I still recommend this paper for publication after major revision, as it may be an inspiration for works on developemnt of more effective methods.

Specific comments:

  1. The major problem with this method is that relatively high intensity of fluorescence was detected in control experiments with non-labeled phages. Even when differences between control and tested samples were statistically significant, they were rarly larger than 2-times. This is a small difference as for fluorescence intensity which is also evident when analyzing microscopic pictures. It is really hard to assess efficiency of phage penetration to different organs at such a high backgroud level of the signal. Surprisingly, this problem is not even mentioned in Discussion, while it should be extensively analyzed and discussed.
  2. Along to my criticism described in p.1, there is no speculation about the source of the high fluorescence background in the experiments with mouse organs. Inetestingly, analogous backgroud in experiments with cultured macrophages was very low. Tha authors should deeply discuss this difference and possible reasons for it.
  3. It is not possible to assess whether the high backgroud signals come from un-labeled phages (very unlikely, though not impossible) or from animal tissue (more likely). Ideally, control experiments should be performed in which mice are treated with a buffer insttead of phages. If no signal is detected, the backgroud comes from un-labeled phages, if the signal is still the same, animal tissues are fluorescent to some extent. I expect the authors might already had such data in their hands. If so, they should be demonstrated.
  4. Another problem with this system is that RFP is a relatively large protein. Could it influence distribution of bacteriophages in the mouse organism? This should be discussed.
  5. There are problems with interpretations of many results. In results demonstrated in Fig. 8, statistically significant differece between fluorescence of control and labeled phage is demonstrated only for muscles. However, kidneys are also mentioned in the text as organs in which phages were detected. In the absence of significant differences, such a conclusion cannot be drawn.
  6. The same problem appears in interpretation of results presented in Fig. 9. Statistically significant differeces between fluorescence of controls and labeled phages are indicated only for lymph nodes and spleens, while in the text, detection of phages in lungs, kidneys and hears is described. This is a wrong interpretation of the results. A lack of statistical significance means that there are no reals differences between control and tested samples, and determined various values represent only a random fluctuation between particular measurements. 
  7. In legends to Figures 1, 5, 7, 8, 9, 10, and 11, there is no information about number of repetitions of experiments. Moreover, it is not indicated what do error bars mean (SD, SE or other parameter)?
  8.  Figure 5 should include statistical analyses not only A vs. control, but also B vs. control, and A vs. B.
  9. Names of genes should be written in italic font throughout the text.
  10. It is stated that purification of labeled phages was conducted as described by Boratynski et al., but this paper is neither cited in the text (no reference number is given) nor mentioned in the reference list.
  11. Results cited in the first paragraph of Results (RFP-Hoc fusion protein production assessed by SDS-PAGE) as "data not shown" should be demonstrated as a figure, either in the main body of the text or in supplementary material.  

Round 2

Reviewer 1 Report

The authors have satisfactorily answered my questions and made significant changes to the revised version.  
Thus, I would recommend the acceptance of the revised manuscript.

Reviewer 2 Report

The authors responded to my previous comments and improved the manuscript accordingly. I do not have further comments.